# Mechanisms Underlying Muscle-Related Diseases and Aging: Insights into Pathophysiology and Therapeutic Strategies

**DOI:** 10.3390/muscles4030026

**Published:** 2025-07-31

**Authors:** Jialin Fan, Zara Khanzada, Yunpeng Xu

**Affiliations:** 1School of Arts and Sciences, Chemistry and Chemical Biology Cancer Institute of New Jersey (CINJ), Director’s Office Cancer Institute of New Jersey (CINJ), Rutgers University, New Brunswick, NJ 08901, USA; 2Department of Molecular Biology and Biochemistry, Rutgers University, Piscataway, NJ 08854, USA; zrk9@scarletmail.rutgers.edu

**Keywords:** muscle aging, mitochondrial dysfunction, muscle stem cells, neuromuscular junction, inflammation, therapy

## Abstract

Skeletal muscle aging and related diseases are characterized by progressive loss of muscle mass, strength, and metabolic function. Central to these processes is mitochondrial dysfunction, which impairs energy metabolism, redox homeostasis, and proteostasis. In addition, non-mitochondrial factors such as muscle stem cell exhaustion, neuromuscular junction remodeling, and chronic inflammation also contribute significantly to muscle degeneration. This review integrates recent advances in understanding mitochondrial and non-mitochondrial mechanisms underlying muscle aging and disease. Additionally, we discuss emerging therapeutic approaches targeting these pathways to preserve muscle health and promote healthy aging.

## 1. Introduction

Skeletal muscle, constituting approximately 40% of total body mass, plays a vital role in locomotion, metabolic homeostasis, and overall systemic health [1,2,3,4]. During aging and in the context of chronic diseases, skeletal muscle experiences progressive degenerative changes, commonly manifested as sarcopenia, frailty, and metabolic decline [5]. These changes are not merely cosmetic or functional—they profoundly impact quality of life and are tightly associated with morbidity and mortality in the elderly.

### 1.1. Evolving Definitions, Burden, and Heterogeneity

Diagnostic criteria for sarcopenia have matured, moving from a focus on low lean mass to multidimensional constructs that incorporate muscle strength, physical performance, and increasingly muscle quality indices derived from imaging and molecular profiling. Consensus groups now emphasize strength as the primary indicator, while body composition and performance refine staging and prognosis. Yet prevalence estimates still vary widely depending on cut-points, ethnicity, and assessment modality. Recent single-cell and spatial multi-omics atlases of human skeletal muscle across the adult lifespan reveal extensive interindividual heterogeneity in fiber type composition, satellite cell pools, immune infiltrates, fibro-adipogenic progenitors (FAPs), and neuromuscular junction (NMJ) signatures that likely underlie variable clinical phenotypes [6].

### 1.2. Hallmarks of Muscle Aging: Beyond Atrophy

Among the cellular hallmarks of muscle aging and diseases, mitochondrial dysfunction has been extensively studied and recognized as a central contributor [7,8,9,10,11,12,13,14]. Altered mitochondrial dynamics, bioenergetic deficits, and increased oxidative stress are tightly linked to muscle atrophy and decreased regenerative capacity. However, emerging research has broadened our understanding of muscle aging beyond mitochondria alone. Increasing evidence now points to a network of interrelated mechanisms, including muscle stem cell (satellite cell) dysfunction, neuromuscular junction (NMJ) instability, and chronic low-grade inflammaging (also known as inflammation)—all of which contribute to the initiation and progression of muscle deterioration with age and muscle related diseases [13,15,16,17,18,19].

### 1.3. Interconnected Pathophysiology: A Network Model

Importantly, these processes do not act in isolation. Mitochondrial dysfunction can impair stem cell activation and regeneration [20,21,22,23], while NMJ disruption may result from and exacerbate mitochondrial, inflammatory damage or muscle stem cell dysfunction [24,25,26,27,28,29]. Inflammation further creates a hostile tissue environment that alters muscle stem cell fate, accelerates NMJ degeneration, and impairs mitochondrial maintenance [30,31,32,33,34,35]. Together, these intertwined factors form a complex pathophysiological network driving age-associated muscle decline.

### 1.4. Therapeutic Landscape: Converging and Diverging Pathways

Given the networked nature of muscle aging, combinatorial interventions are increasingly advocated. Key strategy classes include:

Lifestyle Foundations—Progressive resistance training (RT) remains the cornerstone of sarcopenia prevention, as it robustly improves muscle strength, mass, and functional outcomes. Synergistic benefits arise when RT is combined with aerobic or balance training and supported by optimized nutrition [36]. Caloric restriction, when carefully implemented, enhances insulin sensitivity and metabolic health but risks lean mass loss if performed in isolation [37]. Exercise acts as a potent amplifier of these dietary strategies by enhancing mTORC1 (mammalian Target of Rapamycin Complex 1) signaling, mitochondrial biogenesis, and neuromuscular stability, highlighting the value of combined, personalized lifestyle interventions [38].

Inflammation-Directed Therapies—Strategies range from lifestyle (exercise, diet quality, weight management) that reduce systemic inflammatory load to emerging pharmacological and cell-based immunomodulators targeting cytokine axes, gut–muscle interactions, or senescent immune cells [39,40].

Stem Cell and Regenerative Niches—Efforts to rejuvenate MuSC number and function include metabolic reprogramming, niche ECM (Extracellular Matrix) engineering, modulation of Notch/Wnt/TGF-β signaling, and ex vivo expanded or gene-edited cell transplantation. Clearing senescent niche cells or SASP (Senescence-Associated Secretory Phenotype) factors may enhance engraftment and regeneration [41,42].

### 1.5. Scope and Structure of This Review

In this review, we categorize these mechanisms into mitochondria-dependent and mitochondria-independent pathways, emphasizing how they intersect to influence muscle health in aging and disease. We also summarize recent advances in therapeutic interventions, including pharmacological, genetic, and lifestyle-based strategies, that target these distinct yet interconnected pathways to restore muscle function and resilience.

## 2. Mitochondrial Mechanisms in Muscle Aging and Disease

### 2.1. Mitochondrial Quality Control (MQC) System

Mitochondria play a central role in energy production within skeletal muscle, meeting the high ATP (Adenosine Triphosphate) demands required for contractile activity and metabolic regulation [34,35]. The maintenance of mitochondrial health relies on the mitochondrial quality control (MQC) system, which integrates several interconnected processes: mitochondrial biogenesis, dynamics (fusion and fission), mitophagy, and proteostasis [43,44,45]. Together, these mechanisms preserve mitochondrial integrity and function (Figure 1).

#### 2.1.1. Mitochondrial Biogenesis

Mitochondrial biogenesis is the coordinated process by which cells generate new mitochondria to meet energetic and metabolic demands. This process is primarily governed by PGC-1α (Peroxisome Proliferator Activated Receptor Gamma Coactivator 1a), a master regulator of mitochondrial transcriptional programs. Upstream signaling pathways, including AMPK (AMP Activated Protein Kinase), SIRT1 (Sirtuin 1), and mTOR (Mechanistic Target of Rapamycin), converge on PGC-1α (Peroxisome Proliferator Activated Receptor Gamma Coactivator 1a) to promote the expression of nuclear and mitochondrial genes involved in mitochondrial replication, transcription, and protein import. Through this network, cells dynamically adjust mitochondrial content and function in response to physiological stimuli such as energy stress, exercise, and nutrient availability [46,47,48,49,50,51,52,53,54,55,56,57,58].

#### 2.1.2. Mitochondrial Dynamics

Mitochondrial dynamics involve the tightly regulated processes of fusion and fission that continuously remodel the mitochondrial network to maintain organelle integrity, distribution, and function. Fusion, primarily mediated by the MFN1 (Mitofusin 1) and MFN2 (Mitofusin 2) on the outer mitochondrial membrane (OMM), together with OPA1 (Optic Atrophy 1) on the inner membrane (IMM), facilitates the merging of mitochondrial membranes, promoting mitochondrial content mixing and functional complementation. Conversely, fission, largely orchestrated by the dynamin-related protein DRP1 (Dynamin-related protein 1), enables mitochondrial division, which is critical for mitochondrial biogenesis, distribution during cell division, and the segregation of damaged mitochondria earmarked for degradation. The dynamic balance between fusion and fission is essential for preserving mitochondrial morphology, quality control, and cellular homeostasis [10,59,60,61,62].

#### 2.1.3. Mitophagy

Selective autophagic removal of dysfunctional mitochondria is orchestrated by pathways including PINK1 (PTEN-Induced Putative Kinase 1)/Parkin and receptor-mediated mechanisms involving BNIP3 (Bcl-2/adenovirus E1B 19 kDa-interacting protein 3), NIX (Bcl-2/adenovirus E1B 19 kDa-interacting protein 3-like), and FUNDC1 (FUN14 domain-containing protein 1) [63,64,65,66]. Proteostasis: Mitochondrial proteases like LONP1 (Lon Peptidase 1) and the CLPX-CLPP complex (Caseinolytic Mitochondrial Matrix Peptidase Chaperone Subunit, Caseinolytic Mitochondrial Matrix Peptidase Proteolytic Subunit) degrade oxidatively damaged proteins, preventing their accumulation [67,68,69,70,71]. These proteolytic systems work alongside the ubiquitin–proteasome and lysosomal pathways to maintain proteostasis. Mitochondria-Derived Vesicles (MDVs): In response to mild mitochondrial stress, MDVs bud off selectively from mitochondria to transport damaged proteins or lipids to lysosomes or peroxisomes for degradation. Unlike mitophagy, MDVs allow for compartmentalized quality control without the need for wholesale organelle removal [72,73,74,75,76,77].

#### 2.1.4. Exophers

In post-mitotic cells, including neurons and muscle fibers, exophers have recently been identified as a novel mechanism for the selective extrusion of damaged mitochondria and aggregated proteins. These large, vesicle-like structures provide an auxiliary quality control pathway that complements classical autophagy, particularly under conditions where autophagic capacity is overwhelmed or impaired. By facilitating the removal of deleterious cellular components, exophers contribute to maintaining cellular homeostasis and proteostasis [81,82,83].

Most components of MQC (mitochondrial quality control) decline with aging [84,85]. Reduced biogenesis limits mitochondrial renewal, while imbalanced fusion and fission disrupt mitochondrial morphology [78]. Impaired mitophagy and proteostasis allow damaged mitochondria to accumulate, leading to increased reactive oxygen species (ROS) production and cellular stress [79,80]. This creates a vicious cycle of mitochondrial dysfunction and muscle degeneration.

### 2.2. Mitochondrial Alterations Induced by Aging

Aging triggers a series of detrimental alterations in mitochondrial function and homeostasis that are closely associated with skeletal muscle degeneration (Figure 2). These changes include impaired mitochondrial protein synthesis and clearance, as well as mitochondrial damage caused by reactive oxygen species (ROS) and genetic mutations [19,61,86].

During aging, decreased activity of (AMP Activated Protein Kinase), SIRT1 (Sirtuin 1), and PGC-1α (Peroxisome Proliferator Activated Receptor Gamma Coactivator 1a), along with increased mTOR (Mechanistic Target of Rapamycin) signaling, leads to reduced mitochondrial protein synthesis. Additionally, dysfunction of the proteasome, autophagy–lysosome system, mitochondrial-derived vesicles (MDVs), exophers, and mitochondrial dynamics (fusion and fission) collectively impairs the clearance of damaged mitochondrial components [77,82,87,88,89,90]. Moreover, elevated levels of ROS (reactive oxygen species), increased glycation, accumulation of genetic mutations, and reduced antioxidant defenses such as superoxide dismutase (SOD) further compromise mitochondrial integrity and function [91,92,93,94,95]. These age-related disruptions—encompassing impaired mitochondrial biogenesis, altered dynamics, elevated oxidative stress, and maladaptive stress responses [96,97], ultimately contribute to the progressive decline in skeletal muscle function.

#### 2.2.1. Reduced Mitochondrial Biogenesis and Content

Aging is associated with the downregulation of PGC-1α (Peroxisome Proliferator Activated Receptor Gamma Coactivator 1a), along with dysregulation of key upstream signaling pathways such as AMPK (AMP Activated Protein Kinase), SIRT1 (Sirtuin 1), SIRT6 (Sirtuin 6), NAD (Nicotinamide Adenine Dinucleotide), and mTOR (Mechanistic Target of Rapamycin) [46,53,98,99,100,101,102,103,104,105,106,107]. These changes lead to decreased mitochondrial content and oxidative capacity. As a result, oxidative phosphorylation becomes less efficient, reducing ATP (Adenosine Triphosphate) production and contributing to muscle fatigue, atrophy, and diminished endurance.

#### 2.2.2. Disrupted Mitochondrial Dynamics

The balance between mitochondrial fusion and fission is critical for maintaining mitochondrial shape and function [108,109]. In aged muscle, this balance is disturbed, often shifting toward excessive fragmentation or abnormal hyper-fusion. Both conditions impair mitochondrial distribution and energy output, exacerbating muscle aging and the development of muscle-related diseases [110,111].

#### 2.2.3. Increased Oxidative Stress

Aging reduces the efficiency of the electron transport chain and weakens mitochondrial antioxidant defenses, including enzymes such as SOD2 (Superoxide Dismutase 2), catalase, and glutathione peroxidase [112,113,114,115,116,117,118,119]. This results in elevated production of mitochondrial reactive oxygen species (ROS), which damage mitochondrial lipids, proteins, and mtDNA. The accumulation of oxidative damage further compromises mitochondrial integrity and function [120,121].

#### 2.2.4. Altered Mitochondrial Unfolded Protein Response (UPR^mt^)

The mitochondrial unfolded protein response serves as a protective mechanism against proteotoxic stress by upregulating chaperones (e.g., HSP60, mtHSP70) and proteases to restore protein homeostasis [122,123,124,125,126]. However, during aging, UPR^mt^ (Mitochondrial Unfolded Protein Response) signaling may become chronically activated or dysregulated, leading to a maladaptive response. Instead of restoring balance, persistent UPR^mt^ can amplify cellular stress and contribute to the development of sarcopenia and mitochondrial myopathies [125,127,128,129].

Together, these mitochondrial alterations culminate in the accumulation of dysfunction. This mitochondrial failure plays a central role in driving muscle degeneration during aging, forming a self-reinforcing cycle that accelerates functional decline.

## 3. Non-Mitochondrial Mechanisms in Muscle Aging and Disease

### 3.1. Muscle Stem Cell Dysfunction

Muscle regeneration is impaired in aging due to reduced function and proliferative capacity of muscle stem cells (MuSCs) [20,21,23,130,131,132,133,134]. Age-related alterations—such as downregulation of the WNT (Wingless Integrated Signaling Pathway) and Notch (Neurogenic locus notch homolog) signaling pathways, decreased activity of AMPK (AMP Activated Protein Kinase), PGC-1α (Peroxisome Proliferator Activated Receptor Gamma Coactivator 1a), mTOR (Mechanistic Target of Rapamycin), and WISP1 (WNT1 Inducible Signaling Pathway Protein 1), along with increased levels of TGF-b (Transforming Growth Factor Beta), and miR-34a—contribute to MuSC dysfunction [132,135,136,137,138,139]. This decline in MuSC function compromises muscle repair and regeneration, thereby promoting the onset and progression of muscle-related diseases. In addition, mitochondrial dysfunction, neuromuscular junction (NMJ) deterioration, and chronic inflammation—common features of aging—further exacerbate MuSC impairment. Conversely, MuSC dysfunction can also feedback to aggravate age associated NMJ degradation, mitochondrial decline, and inflammatory responses, forming a vicious cycle that accelerates muscle aging [21,23,28,30,133,134,140,141,142,143,144] (Figure 3).

### 3.2. Neuromuscular Junction Dysfunction

Neuromuscular junction (NMJ) degeneration and remodeling are hallmarks of aging skeletal muscle and play a central role in the decline of neuromuscular transmission [24,28,140,141,145]. Age-related NMJ dysfunction contributes to progressive muscle weakness and increases the susceptibility to muscle degeneration. With aging, elevated levels of IL-6 (Interleukin 6), NF-κB (Nuclear Factor kappa-light-chain-enhancer of activated B cells), mTOR (Mechanistic Target of Rapamycin), 4EBP1 (Eukaryotic Translation Initiation Factor 4E-Binding Protein 1), and matrix metalloproteinases (MMPs), combined with reduced expression of SIRT1 (Sirtuin 1) and TrkB (Tropomyosin Receptor Kinase B), contribute to both structural disintegration and functional impairment of NMJs [26,48,146,147,148,149,150,151,152] (Figure 3). The resulting loss of NMJ integrity disrupts motor neuron–muscle communication and further accelerates muscle atrophy and the progression of muscle-related diseases. Furthermore, age-associated mitochondrial dysfunction, MuSC (muscle stem cell) impairment, and chronic inflammation can further compromise NMJ structure and function, establishing a reciprocal network of degeneration that drives muscle aging [25,27,28,142,153].

### 3.3. Inflammation and Immune Cell Infiltration

Chronic inflammation (inflammaging), characterized by increased IL-6 (Interleukin 6), NF-κB (Nuclear Factor kappa-light-chain-enhancer of activated B cells), TGF-b (Transforming Growth Factor Beta), TNF-a (Tumor Necrosis Factor Alpha), and CCL2 (C-C Motif Chemokine Ligand 2), along with reduced expression of the anti-inflammatory cytokine IL-10 (Interleukin 10), is a key extrinsic factor in muscle aging [32,154,155,156,157,158,159,160,161,162,163,164,165,166]. Increased immune cell infiltration and sustained cytokine production disrupt muscle homeostasis, accelerate tissue degeneration, and impair the regenerative microenvironment. This inflammatory milieu further compromises muscle stem cell (MuSC) function, neuromuscular junction (NMJ) integrity, and mitochondrial health, thereby exacerbating muscle aging and promoting the development of muscle-related diseases [30,31,167,168].

As summarized in Figure 3, muscle aging is driven by the interplay of mitochondrial dysfunction, chronic inflammation, muscle stem cell (MuSC) impairment, and neuromuscular junction (NMJ) disruption. Mitochondrial damage in aging muscle leads to increased reactive oxygen species (ROS), impaired oxidative phosphorylation, and release of mitochondrial DNA (mtDNA), which activate inflammatory pathways such as NLRP3 and cGAS-STING. This chronic low-grade inflammation further disrupts the muscle microenvironment [169,170,171,172]. Inflammatory mediators (e.g., IL-6, TNF-α, TGF-β, NF-κB) exacerbate mitochondrial dysfunction, impair MuSC regenerative capacity, and contribute to NMJ degeneration. In MuSCs, mitochondrial metabolic decline and altered signaling (e.g., WNT, Notch, AMPK, SIRT1, miR-34a) reduce their ability to repair muscle. Damaged MuSCs fail to support NMJ integrity, leading to synaptic fragmentation, denervation, and impaired neuromuscular transmission [18,30,31,32,33,144,154,167,173,174,175,176,177,178,179].

Together, these interconnected mechanisms create a degenerative cycle that drives muscle atrophy, weakness, and the progression of age-related diseases such as sarcopenia and muscular dystrophies.

## 4. Therapeutic Strategies Targeting Muscle Aging and Muscle-Related Diseases

Muscle aging arises from a complex interplay of metabolic decline, chronic inflammation, impaired stem-cell function, and neuromuscular junction deterioration. Therapeutic approaches can be organized by modality—lifestyle, pharmacological, cellular, and anti-inflammatory—each converging on core regulators (PGC-1α, AMPK, SIRT1, mTOR, Nrf2) to maintain energy homeostasis, redox balance, proteostasis, and regenerative capacity. Figure 3 illustrates how these diverse treatments intersect at common molecular nodes to preserve muscle integrity and delay sarcopenia.

### 4.1. Lifestyle Interventions: Exercise and Dietary Restriction

Lifestyle interventions against muscle aging include both exercise and dietary restriction, each acting on multiple interconnected pathways:

Exercise promotes mitochondrial health, stem-cell function, anti-inflammatory signaling, and neuromuscular stability. By activating AMPK (AMP Activated Protein Kinase), SIRT1 (Sirtuin 1) and PGC-1α (Peroxisome Proliferator Activated Receptor Gamma Coactivator 1 Alpha), it drives mitochondrial biogenesis, mitophagy, and antioxidant defenses, while improving glucose uptake and fatty-acid oxidation to enhance metabolic flexibility. Exercise also stimulates MuSC regeneration through modulation of WNT (Wingless Integrated Signaling Pathway), Notch (Notch Signaling Pathway), IGF-1 (Insulin Like Growth Factor 1), TGF-β (Transforming Growth Factor Beta), MAPK (Mitogen Activated Protein Kinase), adiponectin, laminin, cyclin D1, and CCN2 (Cellular Communication Network Factor 2). Its anti-inflammatory effects involve regulation of IL-6 (Interleukin 6), IL-10 (Interleukin 10), Nrf1/2 (Nuclear Factor Erythroid Related Factor), HSP70 (Heat Shock Protein 70), and adiponectin. Finally, exercise maintains neuromuscular junction integrity via mTOR/4E-BP1, NFATc1 (Nuclear Factor of Activated T Cells, Cytoplasmic 1), laminin and miR-206 [11,16,47,49,50,180,181,182,183,184,185,186,187,188,189,190].

Dietary Restriction similarly targets these same core regulators to preserve muscle function. By increasing the AMP:ATP ratio and NAD^+^ (Nicotinamide Adenine Dinucleotide) levels, DR activates AMPK (AMP Activated Protein Kinase) and SIRT1 (Sirtuin 1), which phosphorylate and deacetylate PGC-1α (Peroxisome Proliferator Activated Receptor Gamma Coactivator 1 Alpha) to up-regulate Nrf1/2 (Nuclear Factor Erythroid Related Factor) for improved respiratory capacity and mitophagy. Nutrient limitation also inhibits mTOR (Mechanistic Target of Rapamycin), to enhance autophagic clearance of damaged mitochondria. Concurrently, DR attenuates inflammation [191,192] by down-regulating NF-κB (Nuclear Factor kappa-light-chain-enhancer of activated B cells)–mediated IL-6 (Interleukin 6) and TNF-α (Tumor Necrosis Factor Alpha) production, sustains the satellite-cell pool via SIRT1 (Sirtuin 1) driven metabolic reprogramming, and preserves neuromuscular junction architecture to prevent denervation [37,118,192,193,194,195,196,197,198,199,200,201,202,203,204,205,206,207,208,209,210,211,212,213,214,215,216,217,218,219,220,221,222,223,224,225,226,227,228,229,230].

Together, exercise and dietary restriction converge on a shared network of regulators—AMPK (AMP Activated Protein Kinase), SIRT1 (Sirtuin 1), PGC-1α (Peroxisome Proliferator Activated Receptor Gamma Coactivator 1 Alpha), mTOR (Mechanistic Target of Rapamycin), and NRF1/2 (Nuclear Factor Erythroid Related Factor 1/2)—to coordinate mitochondrial quality control, inflammation resolution, stem-cell renewal, and synaptic stability.

### 4.2. Pharmacological Strategies Targeting Metabolic and Proteostatic Pathways

Pharmacological interventions for muscle aging encompass compounds that target key metabolic and proteostatic pathways. Sirtuin activators such as resveratrol mimic dietary restriction by activating SIRT1 (Sirtuin 1) and upregulating PGC-1α (Peroxisome Proliferator Activated Receptor Gamma Coactivator 1 Alpha), thereby enhancing mitochondrial health and reducing inflammation [231,232,233,234,235,236,237,238,239,240,241]. mTOR (Mechanistic Target of Rapamycin) inhibitors like rapamycin enhance autophagy and proteostasis, reducing the accumulation of misfolded or aggregated proteins [223,242,243,244,245,246,247,248,249,250,251,252,253,254,255,256]. AMPK (AMP Activated Protein Kinase) agonists, including berberine, nobiletin, and tricin, promote mitochondrial remodeling, attenuate oxidative stress, and suppress pro-inflammatory signaling. Notably, tricin, a plant-derived flavonoid, activates AMPK (AMP Activated Protein Kinase) while inhibiting NF-κB (Nuclear Factor kappa-light-chain-enhancer of activated B cells) signaling, thereby counteracting age-related muscle inflammation and degeneration [156,257,258,259,260,261,262,263,264,265,266,267,268,269,270,271,272]. Oleuropein, a phenolic compound derived from olive leaves, improves muscle function by modulating mitochondrial calcium homeostasis, which optimizes bioenergetics and protects against mitochondrial dysfunction [273].

Energy-supporting agents such as NAD^+^ (Nicotinamide Adenine Dinucleotide) precursors (e.g., nicotinamide riboside, NR) replenish NAD^+^ (Nicotinamide Adenine Dinucleotide) pools to drive sirtuin-dependent repair and mitochondrial respiration [274,275,276], while sarcosine improves the pro-regenerative milieu by enhancing one-carbon metabolism and promoting anti-inflammatory macrophage polarization [216,277,278]. Urolithin A, a natural mitophagy activator, has shown benefits in both preclinical models and clinical trials by improving mitochondrial quality control and muscle endurance [279,280,281,282,283,284,285,286,287]. Similarly, β-hydroxy-β-methyl butyrate (HMB), a leucine metabolite, supports muscle protein synthesis, reduces proteolysis, and preserves muscle mass and strength in aging individuals [288,289,290,291,292,293,294,295,296,297,298,299,300,301]. Lastly, mitochondria-targeted antioxidants like MitoQ and SS-31 scavenge reactive oxygen species, stabilize membrane potential, and preserve cristae architecture [302,303,304,305,306,307,308].

### 4.3. Biological Interventions Targeting Stem Cell Function and Neuromuscular Integrity

Cellular and biological therapies for muscle aging focus on both stem cell rejuvenation and neuromuscular protection. Modulating key signaling pathways such as WNT (Wingless Integrated Signaling Pathway), Notch (Neurogenic locus notch homolog), and WISP1 (WNT1 Inducible Signaling Pathway Protein 1) have been shown to rejuvenate muscle satellite cells and their niche, enhancing their proliferative and differentiation capacities, and thereby improving muscle regeneration and repair. In addition, supplementation with nicotinamide and pyridoxine has been reported to stimulate muscle stem cell activity, further promoting regeneration and functional recovery [309]. For example, inhibition of aberrant WNT signaling reduces fibrosis and restores satellite cell function in aged muscle, while activation of Notch signaling or supplementation with WISP1 promotes regenerative capacity [310,311,312,313]. Notably, exposure to a young systemic environment, such as through parabiosis or young plasma transfusion, has been demonstrated to restore satellite cell function in aged mice, in part through rejuvenation of the Notch (Neurogenic locus notch homolog) signaling axis. Interventions such as Muscle-Specific Kinase (MuSK) agonist antibodies and HDAC4 (Histone Deacetylase 4) inhibitors have shown potential in maintaining NMJ structure and function, while exercise has also been shown to delay NMJ degeneration in aging models [314,315,316,317]. Together, these approaches support the maintenance of muscle strength, coordination, and functional integrity during aging.

We emphasize how each therapeutic strategy converges on shared molecular hubs to mitigate muscle decline. This conceptual framework not only clarifies the underlying logic but also highlights opportunities for combination therapies that synergistically target metabolism, inflammation, and regeneration.

## 5. Conclusions and Future Directions

Skeletal muscle aging results from a complex interplay between mitochondrial and non-mitochondrial processes. Mitochondrial dysfunction—including impaired biogenesis and turnover through proteasomal degradation, lysosomal pathways, mitochondrial-derived vesicles, mitophagy, and exopher-mediated clearance—represents a central axis of cellular decline. Concurrently, satellite cell exhaustion, neuromuscular junction (NMJ) instability, and chronic low-grade inflammation synergistically accelerate muscle loss and functional deterioration.

Despite these advances, several important limitations persist. Current insights are largely derived from animal models and in vitro systems, which may not fully recapitulate the cellular heterogeneity and physiological complexity of human muscle aging. Critical variables such as genetic background, comorbid conditions, and sex-specific responses are insufficiently accounted for in preclinical studies. Moreover, the relative contribution and temporal dynamics of individual degenerative pathways remain poorly defined, obscuring the mechanistic order of events and the timing for effective therapeutic targeting.

While numerous pharmacological and genetic interventions targeting mitochondrial health, stem cell function, or inflammatory signaling have shown promise in isolation, the development of combinatorial therapies is still at an early stage. Challenges include determining optimal target combinations, treatment timing, and the management of potential drug–drug interactions or additive toxicities, particularly in aged and frail populations. Furthermore, the translational pipeline is hindered by the lack of validated, muscle-specific biomarkers for disease progression and therapeutic efficacy, as well as limited long-term safety and bioavailability data for candidate compounds.

Future investigations should adopt integrative, systems-level frameworks—incorporating high-resolution single-cell and spatial transcriptomics, multi-omics integration, and longitudinal human cohorts—to comprehensively map the crosstalk between mitochondrial quality control, stem cell dynamics, and neuromuscular integrity across the aging trajectory.

Clinically, this knowledge offers promising translational opportunities. Mitochondria-targeted therapies—such as agents that enhance mitophagy, redox balance, or mitochondrial biogenesis—could be combined with interventions aimed at restoring stem cell regenerative capacity and preserving NMJ integrity. Such multi-pronged strategies may pave the way for precision medicine approaches to delay sarcopenia progression, improve mobility, and extend health-span in aging populations. Furthermore, identifying reliable biomarkers reflecting mitochondrial health or muscle regenerative potential will be critical for early diagnosis, patient stratification, and therapeutic monitoring in clinical trials.

## Figures and Tables

**Figure 1 muscles-04-00026-f001:**
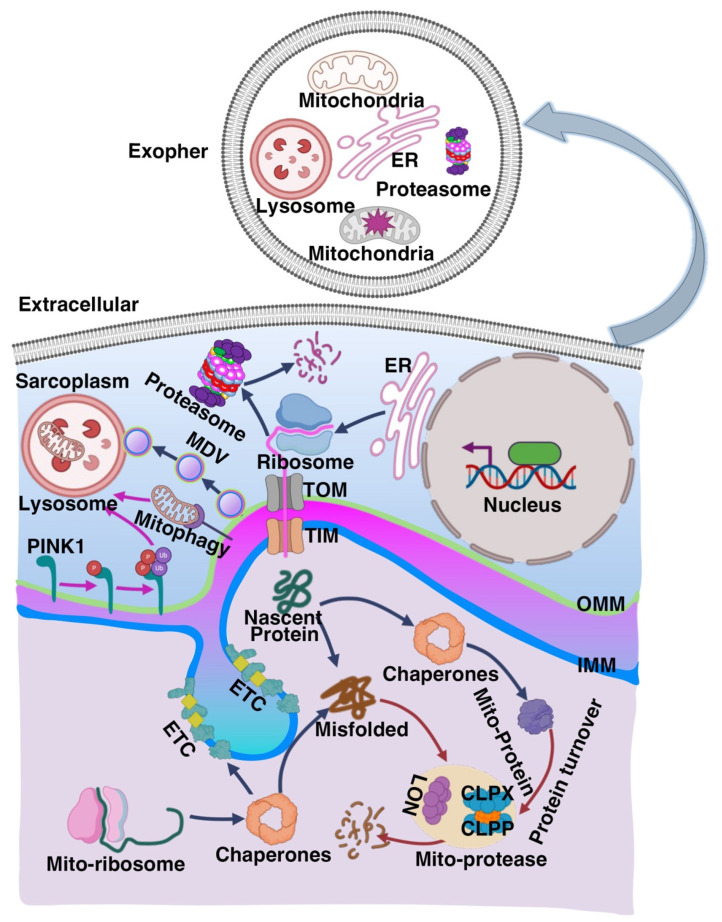
**Overview of the Mitochondrial Protein Quality Control System. Abbreviation:** ATP, Adenosine Triphosphate; CLPP, Caseinolytic Mitochondrial Matrix Peptidase Proteolytic Subunit; CLPX, Caseinolytic Mitochondrial Matrix Peptidase Chaperone Subunit; ER, Endoplasmic Reticulum; ETC, Electron Transport Chain; IMM, Inner Mitochondrial Membrane; LON, LON Protease; MDV, Mitochondrial-Derived Vesicle; Mito-ribosome, Mitochondrial Ribosome; OMM, Outer Mitochondrial Membrane; PINK1, PTEN-Induced Putative Kinase 1; TIM, Translocase of the Inner Membrane; TOM, Translocase of the Outer Membrane [43,44,45,46,47,48,49,50,51,52,53,54,55,56,57,58,59,60,61,62,63,64,65,66,67,68,69,70,71,72]. Mitochondria maintain proteostasis through a series of coordinated mechanisms, including the proteasome, autophagy, lysosomal degradation, mitophagy, mitochondrial-derived vesicles (MDVs), and the exopher pathway. Within mitochondria, nascent proteins are synthesized by the mitochondrial ribosome (Mito-ribosome) and folded with the assistance of molecular chaperones. Misfolded or damaged proteins are degraded by mitochondrial proteases such as LON and the CLPP–CLPX complex or exported for cytosolic degradation. The PQC system also involves the import of nuclear-encoded proteins via the translocase of the outer (TOM) and inner (TIM) mitochondrial membranes. Damaged or dysfunctional mitochondria are removed by mitophagy, a process regulated by PINK1 and lysosomal machinery. Alternatively, MDVs bud from mitochondria to transport damaged contents to lysosomes [42,43,44,45,46,47,48,49,50,51,52,53,54,55,56,57,58,59,60,61,62,63,64,65,66,67,68,69,70,71,72,73,74,75,76,77]. The exopher pathway facilitates the extrusion of damaged organelles and protein aggregates for extracellular clearance [78,79,80]. Together, these pathways coordinate mitochondrial protein turnover and help preserve mitochondrial integrity and function.

**Figure 2 muscles-04-00026-f002:**
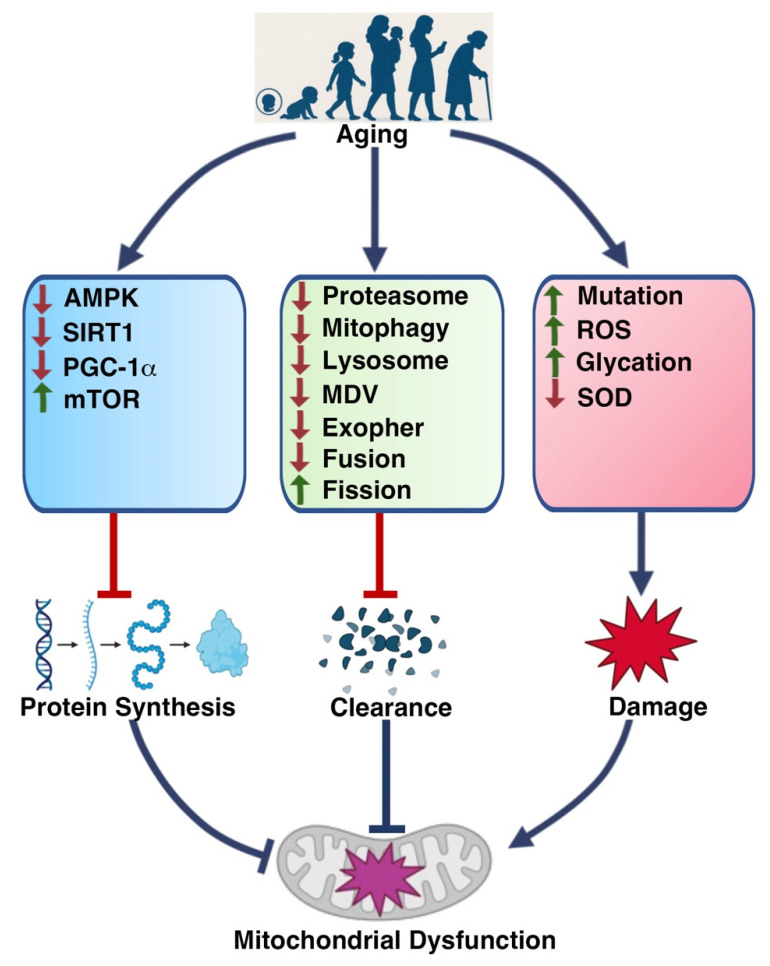
**Relationship Between Aging and Mitochondrial Dysfunction in Skeletal Muscle. Abbreviation:** AMPK: AMP Activated Protein Kinase, ATP: Adenosine Triphosphate, MDV: Mitochondrial Derived Vesicle, mTOR: Mechanistic Target of Rapamycin, PGC1α: Peroxisome Proliferator Activated Receptor Gamma Coactivator 1 Alpha, ROS: Reactive Oxygen Species, SIRT1: Sirtuin 1, SOD: Superoxide Dismutase. Figure 2 depicts the multifactorial impact of aging on mitochondrial integrity in skeletal muscle. Aging impairs mitochondrial biogenesis and protein synthesis by downregulating AMPK, SIRT1, and PGC-1α, while upregulating mTOR signaling. Simultaneously, it compromises mitochondrial clearance mechanisms, including proteasome function, mitophagy, lysosomal degradation, MDV formation, exopher-mediated expulsion, and mitochondrial dynamics (fusion/fission), leading to the accumulation of dysfunctional mitochondria. Aging also increases reactive oxygen species (ROS) production, mitochondrial DNA mutations, and protein glycation, while diminishing antioxidant defenses such as SOD. These converging deficits collectively drive mitochondrial dysfunction—a central hallmark of muscle aging [86,87,88,89,90,91,92,93,94,95,96,97,98,99,100,101,102,103,104,105,106,107,108,109,110,111,112,113,114,115,116,117,118,119,120,121,122,123,124,125,126,127,128,129].

**Figure 3 muscles-04-00026-f003:**
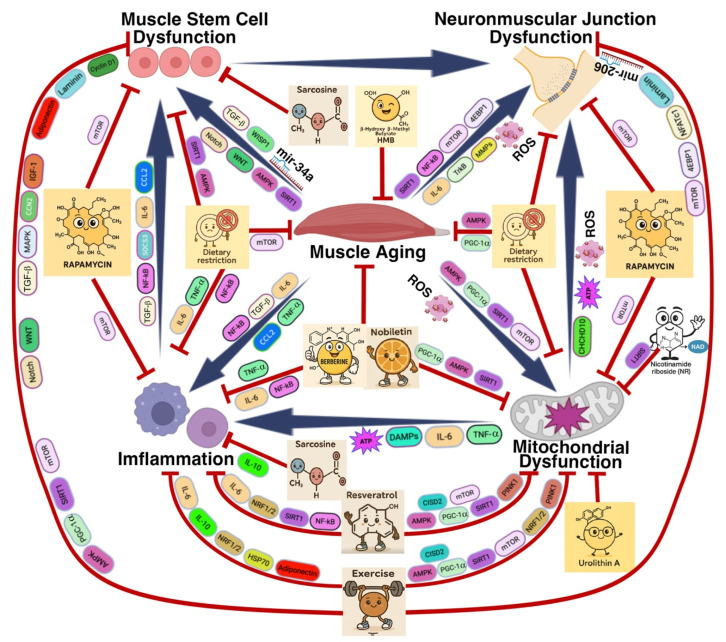
**Mechanisms and Therapeutic Strategies Targeting Muscle Aging and Sarcopenia. Abbreviation:** AMPK: AMP Activated Protein Kinase, ATP: Adenosine Triphosphate, CCL2: C-C Motif Chemokine Ligand 2, CCN2: Cellular Communication Network Factor 2, CHCHD10: Coiled-Coil-Helix-Coiled-Coil-Helix Domain Containing 10, CISD2: CDGSH Iron Sulfur Domain 2, DAMP: Damage Associated Molecular Pattern, HMB: β-hydroxy β-methyl butyrate, HSP70: Heat Shock Protein 70, IGF: Insulin Like Growth Factor, IL-6: Interleukin 6, IL-10: Interleukin 10, MAPK: Mitogen Activated Protein Kinase, MMPs: Matrix Metalloproteinases, miR-206: MicroRNA 206, miR-34a: MicroRNA 34a, mTOR: Mechanistic Target of Rapamycin, NFATc1: Nuclear Factor Of Activated T Cells, Cytoplasmic 1, Notch: Notch Signaling Pathway, NRF1/2: Nuclear Factor Erythroid 1/2 Related Factor 2, PGC1α: Peroxisome Proliferator Activated Receptor Gamma Coactivator 1 Alpha, PINK1: PTEN Induced Putative Kinase 1, ROS: Reactive Oxygen Species, SIRT1: Sirtuin 1, Smad: SMAD Family of Signal Transducers, SOCS3: Suppressor of Cytokine Signaling 3, TGFβ: Transforming Growth Factor Beta, TNF-α: Tumor Necrosis Factor Alpha, TrkB: Tropomyosin Receptor Kinase B, WISP1: WNT1 Inducible Signaling Pathway Protein 1, Wnt: Wingless Integrated Signaling Pathway, 4EBP1: Eukaryotic Translation Initiation Factor 4E-Binding Protein 1. This integrative diagram summarizes the key pathological mechanisms driving muscle aging and sarcopenia, along with corresponding therapeutic interventions. Core drivers include mitochondrial dysfunction, neuromuscular junction (NMJ) deterioration, muscle stem cell (MuSC) exhaustion, and chronic low-grade inflammation (“inflammaging”). Mitochondrial dysfunction leads to excessive ROS production, which damages both NMJs and MuSCs. At the NMJ, oxidative stress and reduced neural input contribute to synaptic degradation and transmission deficits. In MuSCs, disrupted mitochondrial function and aberrant signaling from factors like TGF-β and miR-34a impair self-renewal and regenerative capacity. Inflammaging, mediated by cytokines such as IL-6 and TNF-α, further exacerbates muscle degeneration. Central metabolic regulators—including PGC-1α, AMPK, and SIRT1—play essential roles in maintaining mitochondrial health and cellular energy homeostasis, while NRF1/2 orchestrate antioxidant defenses. Multiple therapeutic interventions target these pathways. Exercise, resveratrol, and nobiletin enhance mitochondrial biogenesis and suppress chronic inflammation. Dietary restriction improves mitochondrial efficiency and dampens systemic inflammatory signaling. Rapamycin mitigates age-related muscle loss by inhibiting mTOR signaling and promoting autophagy. Berberine activates AMPK, reduces oxidative stress, and enhances metabolic function. In addition, β-hydroxy β-methyl butyrate (HMB) supports muscle protein synthesis and reduces proteolysis. Urolithin A promotes mitophagy and improves mitochondrial quality control; and nicotinamide riboside boosts mitochondrial metabolism and supports SIRT1 activity [140,141,142,143,144,145,146,147,148,149,150,151,152,153,154,155,156,157,158,159,160,161,162,163,164,165,166,167,168,169,170,171,172,173,174,175,176,177,178,179]. Collectively, these interconnected mechanisms and interventions offer promising strategies to preserve muscle integrity and counteract sarcopenia.

## Data Availability

The data presented in this study are available on request from the corresponding author.

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
