# Peer review of "Mechanisms Underlying Muscle-Related Diseases and Aging: Insights into Pathophysiology and Therapeutic Strategies"

_muscles, 2025, doi:10.3390/muscles4030026_

Round 1
Reviewer 1 Report
Comments and Suggestions for Authors
This paper reviews the pathophysiology of muscle diseases and the effect of aging in muscles. It provides a summary of the literature on the mitochondrial and non-mitochondrial mechanisms underlying muscle aging and disease. As a review of existing literature, it does not provide novel points beyond what is already known, but presents a comprehensive overview.
Minor suggestions and comments:
- Line 40: replace “inflammaging” with “inflammation”
- Figures: recommend clarifying if this is original work in the legend of the picture
- Line 65 and onwards: please write out abbreviations (even if provided in the legend)
- Line 79 and onwards: recommend breaking up the paragraph of the different components listed into bullet points
- Line 115: replace “this figure” with “figure 2”
- Line 163: replace “dysfunctional” with “dysfunction”
Author Response
We appreciate these helpful suggestions, which have improved the readability of the manuscript. We have revised as your suggestions.
Line 40: Replaced “inflammaging” with “inflammation.”
Figures: Confirmed that all figures are original. We have not explicitly stated this in the legend, as this is generally understood.
Line 65 and onwards: Expanded all abbreviations as recommended.
Line 79 and onwards: Revised the paragraph into bullet points for better clarity.
Line 115: Replaced “this figure” with “Figure 2.”
Line 163: Replaced “dysfunctional” with “dysfunction.”

Reviewer 2 Report
Comments and Suggestions for Authors
The manuscript “Mechanisms Underlying Muscle-Related Diseases and Aging: Insights into Pathophysiology and Therapeutic Strategies” by Fan is a review article which integrates recent advances in understanding mitochondrial and non-mitochondrial mechanisms underlying muscle aging and disease. In addition, the authors discuss emerging therapeutic approaches targeting these pathways to preserve muscle health and promote healthy aging. In general, this article is critical in this field and contains essential contents. I have minor concerns before this manuscript is accepted for publication.
iThenticate report indicates that the percent match is 64%. Therefore, the authors should reduce the similarly.
Author Response
Thank you for your suggestions.
We have substantially revised the text to reduce similarity, including paraphrasing and restructuring sections to ensure originality and clarity.

Reviewer 3 Report
Comments and Suggestions for Authors
In the current review, the authors summarized current knowledge on some of the mechanisms underlying muscle-related diseases and aging.
The review is well-written; nevertheless, some improvements are needed.
The review lists 320 references, many of which are reviews published many years ago. When writing a review, the authors should summarize recent/novel publications and provide their summary of the current state of the subject. It looks like the current review relies heavily on the old, previously published reviews and interpretations. At the same time, some critical references are missing from the statements in the text (see below). My suggestion would be to carefully review the references, remove the ones that are not necessary (old reviews), and add those that are needed. Many of the references have no relationship to muscle, but the statements are extrapolated to processes that occur in muscle.
Suggestions:
Line 98: “Most components of MQC decline with aging. Reduced biogenesis limits mitochondrial renewal, while imbalanced fusion and fission disrupt mitochondrial morphology. Impaired mitophagy and proteostasis allow damaged mitochondria to accumulate, leading to increased reactive oxygen species (ROS) production and cellular stress”. Add references to these statements.
Lines 115-123: Add references to these statements.
Lines 195-216: Add references to these statements.
Carefully review and add references to other statements currently missing in the manuscript.
Add a limitation statement. Expand future directions with a discussion on how this knowledge can be used in the clinic.
Author Response
We are grateful for these valuable suggestions, which have strengthened the scientific rigor and relevance of our review.
References: We have reviewed all 320 references, removed outdated reviews, and incorporated recent and muscle-related studies to ensure that the reference list reflects the current state of the field.
Line 98: Added appropriate references supporting statements about MQC decline, mitochondrial biogenesis, fusion/fission imbalance, and ROS.
Lines 115–123: Added relevant references.
Lines 195–216: Added relevant references.
Other missing references: The manuscript has been thoroughly reviewed to add necessary references to support statements throughout.
Limitations and future directions: We added a limitation statement and expanded the future directions to discuss how this knowledge could be translated into clinical strategies.

Round 2
Reviewer 2 Report
Comments and Suggestions for Authors
The authors addressed all my concerns.
Reviewer 3 Report
Comments and Suggestions for Authors
The authors addressed all of my critiques.